# The Mediation Effect of Media: Artvin, Reverse Migration, and Social Municipalism

Mehmet Kocatepe [1], Cemal Yorgancıoğlu [2,*], Mustafa Sağsan [3] and Harun Şeşen [4]

1 Media and Communication Studies Program, Institute of Graduate Studies, Near East University, Nicosia 99138, North Cyprus, Mersin 10, Turkey; mehmetkocatepe08@gmail.com
2 Department of International Relations, Faculty of Economics, Administrative and Social Sciences, World Peace University, Nicosia 99010, North Cyprus, Mersin 10, Turkey
3 Department of Business Administration, Faculty of Economics, Administrative and Social Sciences, World Peace University, Nicosia 99010, North Cyprus, Mersin 10, Turkey; mustafa.sagsan@wpu.edu.tr
4 Department of Business Administration, Faculty of Economics and Administrative Sciences, Cyprus International University, Nicosia 99258, North Cyprus, Mersin 10, Turkey; hsesen@ciu.edu.tr
* Correspondence: cemal.yorgancioglu@wpu.edu.tr

**Abstract:** Throughout history, migration has had a significant impact on communities, affecting populations, countries, and abandoned or immigrated places in both positive and negative ways. In today's world, it has become a social element with undeniably profound effects on society and individuals. This study aims to explore the impact of municipal services on migration and reverse migration in Artvin Province. Furthermore, this article aims to fill this gap by analysing the mediating role of the media and examining the relationship between social municipalism and reverse migration in Artvin. This article uses the model of deviant case analysis to explain the phenomenon of migration in the case of Artvin. A quantitative approach was adopted and conducted in the provinces (Ankara, Istanbul, Bursa, and Kocaeli) to which people from Artvin have migrated the most. A total of 700 responses were obtained. The results show that there is a positive relationship between social municipalism and migration and that the media has a mediating effect between social municipalism and migration. While traditional media influence people's decision to migrate, social media play an important role in the reverse migration decision.

**Keywords:** migration; reverse migration; social municipalism; social media; traditional media; Artvin Province

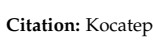



## 1. Introduction

Migration is a phenomenon as old as human history. There is a close relationship between people and the places where they are born and live. The social environment and the physical environment play an important role in the emergence and development of human beings, who occupy a special place among living beings and what is related to human beings [1]. Migration is a concept that can highlight wars, famines, or political unrest and give meaning to international refugee flows. It includes young adults who have to move from one region to another in search of work and middle-aged professionals who seek a retreat to the countryside and then return. It also includes families on the move to meet their housing needs in changing conditions or gipsies and other nomads who have adopted an active lifestyle [2]. The basis of the definition of migration, then, is moving from one geography to another. This event takes place between rural settlements, as well as between urban and rural ones [3].

In the case of Anatolia, migration generally takes place from east to west. The main reason for this migration is the fact that Turkey has a surplus of economic and social resources in the western provinces, as shown by indicators of population and economic activity. Before leaving a place, one undoubtedly gathers a lot of information and evaluates

various factors. Stouffer (1940) makes it clear that pull factors are important for migration to any two centres, but the most important point determining the importance of these factors is distance. It is easier to migrate back to the closer place, but it is much more difficult for those who go far away to come back. In this sense, the difficult option was chosen. Data from the Turkish Statistical Institute (TurkStat or TUIK) cover all 81 provinces since the address-based census began in 2007. Work, employment, health, job changes, marriage, natural disasters, terrorist attacks, religious reasons, land disputes, etc., are some of the major reasons that lead to migration in Turkey [4]. However, the reasons for migration movements in Turkey are also based on economic factors, social well-being, employment opportunities, and municipal factors [5–8].

Another concept that should be studied together with the phenomenon of migration is reverse migration. Reverse migration can be defined as the departure from the city or country to which one has migrated to the old settlement after a certain period of time. Factors that influence the decision to migrate back include gender, education, and income level [9]. In general, these factors are related to the individual's purchasing power in the context of migration, strategic accumulation of human capital, occupational status and social prestige, asymmetric knowledge, social comparison, relative deprivation, marriage or divorce, inheritance, determination to achieve a personal goal, separation costs, and failure [10]. Although there are mass migrations and continuous migrations, each migration is sui generis. In view of these developments and changes in society, it is not enough to consider migration only in a global context. New scientific perspectives and theoretical approaches are needed to understand migration, albeit at the local level.

This article is about the extent to which the activities of municipalities can trigger reverse migration and the extent to which the media (social or traditional) can mediate within the logic of municipalism. This article problematises the link between the social policies of municipalism and reverse migration. Can effective programmes or activities of municipalities be a balm to prevent migration? On the other hand, empirical research in the field of organisational studies shows that social media can improve communication. Recent research on the role of social media in the workplace shows that social media contribute significantly to work performance by enabling immediate effective communication and feedback, which have been shown to be helpful in improving work performance [11]. In this context, the institutional structures of municipalities enable direct communication with citizens. Social media in particular provide a platform for citizens to interact and influence brand communities. The question of why this is important needs to be answered in this sense. Migration, by its very nature, describes change and relocation. This state of transfer (from one place to another) often brings massive problems (otherwise, migration would be unnecessary). Stopping migration, or at least ensuring that it is balanced and predictable, is important for the economy, politics, sociology, and psychology of countries. In this context, the question arises whether good local politics can be a reason to prevent migration. Internal migration can be a serious problem for the life of countries. Could sound policies emerge in an environment where investments are shaped according to needs while populations are constantly on the move? In this sense, the concepts of municipalism and migration have been little explored in the media and generally neglected.

This article uses the model of deviant case analysis to explain the phenomenon of migration in Artvin's case. "Deviant case analysis is the study of particular cases which are anomalous with respect to a given hypothesis" [12] (p. 1). It can be both an exploratory tool and can help to expand what is known about the field without affecting the status quo. Although the relationship between social municipalism and media and the relationship between social media and migration have been discussed in the existing literature, there is a gap that is not emphasised. Therefore, this article aims to fill this gap by analysing the mediating role of media (traditional and social) and exploring the relationship between social municipalism and reverse migration in Artvin. For this purpose, a survey was conducted with the participation of people who migrated from Artvin and now live in different cities in Turkey. Artvin is located on the periphery of Turkey and is one of the



cities furthest from the centre. This green city in the Eastern Black Sea region, located on the border of Turkey with Georgia, is culturally colourful. The population consists of Hamshen, Georgians, Laz, Meskhetian, and Kipchak Turks. It is Turkey's gateway to the Caucasus and has experienced high levels of in-migration. Therefore, it is a unique case study to understand migration and reverse migration phenomena using the deviance analysis model. This article examines the case study of Artvin from the perspectives of the social municipality, migration, and the media.

As for the shortcomings of the study, the authors can point to the lack of female participants (which is a cultural code), since in the Turkish family structure men are generally more likely than women to decide to migrate, and on the other hand, participants with a low level of education did not show enough interest in the survey. A total of 64 questions were asked, avoiding a larger number of questions for fear of distracting respondents.

### 1.1. Migration Overview

The phenomenon of migration, which has affected humanity throughout history, has also contributed to the development of society [13,14]. Migration is an indispensable feature of both the modern world and the traditional world and has undoubtedly transformed societies around the world [15]. It has been observed that countries with declining birth rates and ageing populations tend to absorb immigration to solve demographic problems. Concepts such as cross-border and circular migration preserve the basic ties of migrants with their home countries after migration, and some migrants wish to return to their home countries [16]. Migration, usually perceived as a state of permanent settlement from one's motherland/descent to another place, can take different forms and have different demographic impacts. Migration dynamics are multi-layered and only partly shaped by historical ties. In addition to economic factors, migration is also driven by political instability, religious strife, and environmental degradation and sustained by social networks and migration cultures [17]. Although at first glance migration is seen as a displacement movement, it causes far-reaching changes in individuals and society in terms of its causes and consequences [18]. In this context, it is necessary to define the concept of migration in the relevant literature and to delineate its theoretical boundaries.

To outline the study of migration in general, the work of Ravenstein (based on the phenomenon of industrialisation and urbanisation) is a premise [19] (p. 47), [3] (pp. 198–199). Lee, on the other hand, focused more on migration than on migrants but noted that migrants should not be ignored and identified the factors that keep/attract and repel migrants (push and pull factors) [2] (pp. 49–50). Petersen (1958) identified five types of migration, taking into account individual and class differences [20] (pp. 258–264), while Stouffer's "theory of intervening opportunities" [21] is a micro-oriented theory that focuses on the migrant as a social actor. Within the structuralist approach, Wallerstein's world systems theory (centre–periphery) [22] and migration systems theory show a historical–political–economic approach to the relationship between migratory movements and countries [19] (p. 36). The network of migrant relations (an intersubjective approach) highlighted by Abadan-Unat is an approach based on interpersonal connections (countries of origin and destination and old and new migrants) between migrants [23]. On the other hand, there is a wide range of economically based theories of migration, such as the macro [24] and micro theories of neoclassical economics [25,26], the New Economy of Migration Theory [27,28], the Segmented Labour Theory [29], and the neo-Marxist development theory within a centre–periphery approach [30]. This article uses Lee's migration theory and contributes to the existing literature in the context of pull and push factors (municipal, reverse migration, traditional media, and the mediating role of social media).

### Reverse Migration

This title usually deals with the regions receiving return migration. It is possible to summarise the causes of reverse migration under several headings. Studies explain reverse migration by reasons such as the development of technology, the emergence of new growth

and employment opportunities in developing places, sustainable growth in agriculture with an increase in rural employment and production, and change and development to increase the income of individuals in unequal economic and political conditions of the country. For example, the reverse migration process in China has been explained by the impact of technological development and has found that technology is an important contributor to the outflow of skilled labour in China [31]. Another study looking at reverse migration explains the migration from developed areas to developing areas, such as the departure of skilled Indian workers from the US to take advantage of new growth and employment opportunities [32]. It has been noted that development in rural areas and a sustainable agricultural economy will, on the contrary, be an attractive factor for migration. These studies state that the decline in urban wages accompanied by an increase in rural employment and income, or the increase in agricultural income accompanied by no change in industrial production, will lead to reverse migration [33,34].

These studies show that there is a relationship between development and migration. Advances in economic and social development have positive and negative impacts on migration. Migration movements are expected to decrease as the standard of living and development of countries, regions, and cities converge. Accordingly, the decrease in the difference between income levels and the higher income level in the original place of residence will reduce migration [35] (pp. 488–489). In this case, the relationship between development and migration is said to be an inverted U-shaped. Wiltshire [36] (pp. 63–64) notes that push factors in particular play a more dominant role in metropolitan areas than pull factors in developing cities. He examines the causes of reverse migration in Japan within the framework of two basic views. The first view refers to the functional growth and rapid development of cities outside metropolitan areas. The second view, on the other hand, states that the relocation of manufacturing from metropolitan areas has caused reverse migration.

Reverse migration, which is mainly dealt with in the literature in the framework of the push–pull theory [37,38], can be considered as a new mobility force that has started in Turkey in recent years. It is possible to explain the reverse migration process in Turkey with many factors. However, examining these factors in the context of push and pull theory provides a better understanding of migration and reverse migration in Turkey [39]. In the 2000s, urban areas in Turkey began to both push and pull. The inflationary economy, which puts a strain on living conditions, accelerates the interurban migration of people. This development implies a different kind of migration from urban areas and shifts the return flows to smaller cities. The unbearable burden of rent, food, and transport in metropolitan areas can push people in this direction. It may be that people hopelessly return to their "destitute" villages or hometowns from the "prosperous" city to which they hopefully immigrated [40]. In this context, in terms of flow direction, it is argued that there is reverse migration from urban areas to developing rural areas or cities in Turkey [41].

### 1.2. Theoretical Linkages of Social Municipalism, Media, and Migration

### 1.2.1. Social Municipalism

Social policy aims to protect the rights of workers. Over time, it has become increasingly necessary to implement reforms that include groups such as the unemployed, youth, children, the elderly, the disabled, and women, which is inevitable. While the needs of these populations could have been met by non-governmental organisations, religious organisations, and philanthropists, due to globalisation, migration, and rapid urbanisation [42] (p. 9), the basic needs of these populations could not be met by these organisations. Under these circumstances, municipalities have had to take over the task of social policy. States have adopted a less interventionist approach, and the increase in these aforementioned people has created a large gap. To respond to this problem, municipalities have tried to fill this gap to a large extent.

Social municipalism entrusts municipalities with the task of planning and regulating the field of social policy. In this context, social municipalism is an understanding that aligns

housing, health, education, and the environment with a social purpose by including them in public spending. The goals of social municipalism can be summarised as: ensuring social solidarity and integration, providing the necessary infrastructural investment for social and cultural activities, and helping to increase the dwindling elements of social security and equity between the individual and society [43] (p. 36).

Local governments in Turkey consist of three units: the provincial special administration, the municipality, and the village administration. Social policy activities for aforementioned communities are the main task of all local government units. Social municipalism is the preferred concept to describe local government units [44] (p. 457). Since the Second World War, municipalities in Turkey have risen to a much more important position than the special provincial administrations. Rapid population growth and migration movements have led to the emergence of unplanned cities, resulting in an increase in demand for urban services. The area served by municipalities is larger, and the resources they use are more extensive. Municipalities, which carry out their functions through statutory schemes, have assumed an important role in planning, implementing, and evaluating the implementation of social policies [45] (pp. 17–18).

Rather than being an alternative to capitalism, social municipalism in Turkey aims to repair the disruption and destruction caused by capitalism (to some extent) and make the system sustainable. Local welfare, which is distributed at the nepotic level, is far from being systematic and rights-based. Social municipalism, which partially meets the need-based social needs of society, also contributes a little to the understanding of social justice by distributing some of the profits to the central and local authorities. These services, far from being a systematic, general social practice with certain principles and standards, still provide a livelihood for the poor section of society in cities. The social stratification that has become more evident through migration has further increased urban poverty. To counteract these problems, social municipalism has been put into practice. Through these practices, municipalities are also able to mobilise civil and informal resources [46].

### 1.2.2. Social Municipalism and Media Relations

Municipalities always pursue the goal of communicating their social practices to their target audiences via media. In this context, social media are an indispensable tool for public communication with citizens. Social media can help organisations receive more detailed and useful feedback [47] (p. 1). Karakoç stated that the media has an accelerating effect on the spread of popular culture and that the media has an influence on human decisions [48] (p. 3).

The use of social media from the mid-2000s and onwards has led to the rise of Big Data and the global integration of communities around the globe [49]. From this point of view, the journey of the web can be described as a network of knowledge, communication, cooperation, and integration. Two-way communication between people and authorities ensures greater engagement between locals and municipalities and can be seen as a win–win. On the one hand, it allows municipalities to take more comfortable and healthy steps towards institutionalisation, and on the other hand, it increases the use of social media by citizens, which contributes to political participation. This should be briefly considered as a matter of dialogue [48]. In this sense, it is a medium where a citizen can express his grievances succinctly and receive a solution to his problem that requires institutional transparency, and the solution can be heard by all [47].

New media platforms offer benefits not only for the services provided by municipalities but also for communicative needs, democracy, and political participation [50] (p. 68). New media can be used to enhance the free flow of information and expression, enriching socio-political discussions [51] (p. 124). They can be used as a real-time communication channel that provides instant information. Thus, they can also be used strategically in the management of crisis or disaster situations. In addition to these advantages, new media also pose risks for municipalities, such as privacy and protection of personal data [52] (p. 206). In addition, the design of websites that are more for promotion than interaction, lack of

updates, bureaucratic obstacles, and insufficient technical staff remain the main challenges in managing the use of social media by municipalities [50] (p. 75); [53] (pp. 77–78).

### 1.2.3. Social Media and Migration

Over time, social structure has developed in parallel with technology. Accordingly, the evolution from oral culture to digital culture has become the expression of changes in communication technologies. Society (as a structure) cannot be separated from communication, which enables the transmission of value systems between people. Social change is primarily related to the differentiation of communication patterns, changing systems of thought and meaning, and thus to changes in the way individuals live and in their culture. In this context, the interrelationship between migration and technology and the changes they bring about lead to major upheavals and cultural revolutions. The most obvious example of how migration and technology together have changed society is the Industrial Revolution. Castells [54] argues that new communication technologies have had the same effect as the Industrial Revolution, changing the social structure by affecting the form of relationships and interactions in society. From this point of view, it is necessary to examine the phenomenon of migration as a secondary factor influencing the formation of digital society through technology.

One of the greatest advances of social media is the ability to act collectively. The organisation of social movements in the network society takes place thanks to the Internet [55] (pp. 138–139). In general, social movements driven by new communication technologies represent scenes of collective action [56] (p. 62). Virtual meeting places provide social networks and encourage online protests that gradually contribute to the strengthening of democracy [57] (p. 201). The journey of the network society from concrete space to virtual space is not only a form of displacement that can be expressed as mobilisation but also a multidimensional migration movement. It prepares the ground for the formation of a new kind of culture and society with the differentiation of forms of communication and interaction. In this context, "migration", "social media", and "social municipalism", which are the focus of social sciences, can be important analytical tools to understand the social change that accompanies digitalisation and contributes to solving problems. To this end, this study aims to shed light on understanding the complex relationship between media and migration studies that could ultimately help address the social problems associated with migration/return.

### 1.2.4. Migration and Social Municipalism

Global practice shows that social policy at the local level can be managed within the framework of social municipalism, which allows municipalities to be interpreted as local agents of the welfare state [58]. Municipalities, which in the past were seen as complementary to central government, have increased their effectiveness in social welfare and general social policy, especially in the wake of the globalisation process [59] (p. 188). Although municipalities are responsible for policies within their geographical boundaries, these policies are not independent of government oversight and market conditions. There is a legal framework for municipalities' jurisdiction, and municipalities' budgets are set accordingly. However, this legal and economic framework cannot lead to a subordinate role for municipalities in the field of social policy. Social policy aims to minimise social problems by assessing them from a comprehensive perspective. In this context, social municipalism, which reduces social policy to the local level, can also be called a mirror of the welfare state at the local level [60].

Whether they are migrants, asylum seekers, or refugees staying at their borders, municipalities are primarily concerned with keeping the necessary records and providing short-term solutions to the social needs of these people. In particular, the growing foreign population has increased the responsibility of central governments, municipalities, and NGOs [61]. In this sense, the vulnerability of migrants in the economic and social spheres is likely to have a direct and profound impact on the practices of municipalities in the regions

where they live [62]. In this context, the role of municipalities in housing, employment, health, education, and democratic participation in integration policies to be implemented based on the basic needs of migrants will be undeniably important [63].

1.2.5. Artvin, Migration, and Social Municipalism

Given today's developments, Turkey is at the centre of migration research as a country that is both a source, destination, and transit country for various migrations. Tracing the historical line of development of migratory movements in Turkey, it is noticeable that the most common and predominant migrations are rural–urban, urban–urban, and international migrations to Turkey. Especially in the era of globalisation, increasing migratory movements inevitably lead to various problems for host societies. These migrations directly affect the social structure of Turkey and bring different outcomes and problems.

Artvin occupies a different position within the migratory movements in Turkey. This is because, in terms of migratory movements, Artvin is among the five provinces with the highest emigration rate in Turkey. Artvin, which is one of the emigrating provinces, is also located in the destination region of international migration. It would not be difficult to call Artvin, which is located at the crossroads of internal and external migration, a city of migration. The migration experience of Artvin, which lies at the centre of different types of migration and different migration patterns, deserves to be the subject of a sociological study. Indeed, this is the main motivation for this study.

The subject of this study is whether there is a relationship between Artvin's migration experience and social municipal services. In this study, insiders refer to people who have moved from Artvin to big cities, especially Istanbul, Ankara, Bursa, and Kocaeli. On the other hand, outsiders, i.e., those who migrated to Artvin from countries of the former Soviet Union, especially Georgia, are excluded. This study sought to identify the reasons for migration of those who migrated from Artvin and the factors considered by those considering reverse migration. Attempts were made to determine the extent to which the evolving concept of social municipalism has gained prominence and the channels through which its visibility in the media has been determined. In the context of this issue, in order to understand the migratory experience of Artvin, the perspectives of Artvin people living outside Artvin were consulted.

Some of the practices and projects of social municipalism of the municipality of Artvin in this context are:

- 　　Home care services.
- 　　Soup kitchen project.
- 　　Centre for barrier-free living.
- 　　Funeral services.
- 　　Artvin Kabir Project.
- 　　Welcome Baby Project.
- 　　Ladies Club.
- 　　Ramadan Tents.
- 　　Compassion Hand Store.

In addition, there are such practices as strengthening unity and solidarity through events on religious and special days, enriching the city in the social and cultural spheres through sporting events, conferences and seminars, children's playgrounds, walking trails and social facilities, etc. The city centre consists of seven neighbourhoods. The municipality has set up neighbourhood assemblies in each neighbourhood, including the neighbourhood head. With this team of seven, regular information meetings are held in each neighbourhood, attended by all residents of the neighbourhood, to inform and communicate about the problems of the neighbourhood and the work of the municipality. Every month, at least three–four different news items are broadcast on national television. The municipality is in contact with its supporters in many countries around the world through social media, especially Facebook, Twitter, and its website.

## 2. Research Model and Hypotheses

This study explores the traditional and social-media-conditioned relationship between social municipal activities and reverse migration, as shown in Figure 1. Social municipal activities have been preferred in today's world due to the ease of Internet-based interaction with city dwellers. People choose to migrate for political, economic, educational, health, terrorist, and war reasons. The place of migration is chosen to fulfil the reasons for migration, to increase level of wealth, and to ensure a comfortable life [15]. It is envisaged that the policy of a welfare state is to increase the welfare level of the individual. The welfare state plans the overall budget and provides the means for the implementation of welfare state policies to enable people to live in prosperity [64]. The concept of social municipalism is the reflection of the welfare state in municipalities [65]. Municipalities share their social practices with citizens through the media [47]. In this sense, municipalities implement social policies to make their locations a centre of attraction. The social policies implemented vary depending on the settlement of the municipality and the demands of the citizens, which range from bicycle lanes to climate change measures [66–68]. While services offered as welfare state policies are announced to citizens by municipalities and public institutions, social media are widely used [69]. However, in the periphery, traditional media are still considered the main means of communication.

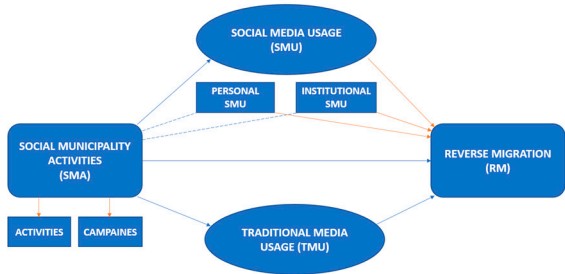

**Figure 1.** Research model.

In this context, the following hypotheses were developed to shed light on the research. The research model of this study is based on the following variables: the dependent variable is reverse migration (RM), the independent variable is social municipality activities (SMAs), and the mediating variables are social media use (SMU) and traditional media use (TMU).

**H1:** *SMA has a positive impact on SMU.*

**H1a:** *SMA has a positive impact on personal SMU.*

**H1b:** *SMA has a positive impact on institutional SMU.*

**H2:** *SMA has a positive impact on TMU.*

**H3:** *SMA has a positive impact on RM.*

**H4:** *SMU has a positive impact on RM.*

**H5:** *TMU has a positive impact on RM.*

**H6:** *SMU plays a mediating role between SMA and RM.*

**H6a:** *Personal SMU plays a mediating role between SMA and RM.*

**H6b:** *Institutional SMU plays a mediating role between SMA and RM.*

**H7:** *TMU play a mediating role between SMA and RM.*

## 3. Methods

A quantitative approach was adopted to understand the relationship between the variables in the research model. A questionnaire with a five-point Likert scale was designed

and applied to people that have lived in Artvin but are currently living in Istanbul, Bursa, Ankara, and Kocaeli. Google Forms was used as a survey tool that allowed easier access to the results, which were analysed numerically. Due to the demographic characteristics of the participants, the survey had to be conducted as a cluster sampling. For the cluster sample, four provinces were selected where Artvin people migrate to intensively. Artvin residents living in these provinces were randomly selected. The participants were randomly approached through associations of compatriots living in the concerned provinces, and the questionnaires were filled out through the Internet. Since this is a method that examines the existence of a relationship between two variables or between several variables, the correlation method was used in this study [70] (p. 132). Survey was carried out between 15 August and 13 September 2019. The questionnaire was distributed to participants via the Internet, and 700 responses were collected. The model created in the study was analysed using structural equation modelling (SEM) via the programme AMOS 16.0.

### 3.1. Demographic Characteristics of the Sample

The Yamane [71] formula was used to calculate the sample size, which resulted in a sample size of 399 residents. Due to potential non-suitable responses, a total of 740 surveys were distributed in four provinces, and 700 responses were collected. A total of 320 respondents from Istanbul, 156 from Bursa, 123 from Ankara, and 101 from Kocaeli participated in the survey, and respondents were asked a total of 64 questions. Table 1 shows the demographic characteristics of the sample in the four provinces. The population under research comprises 279,243 individuals who originally hail from Artvin in Turkey but are presently residing outside the province. The survey was executed in four cities— Ankara, Istanbul, Bursa, and Kocaeli—which account for 71.2% of the estimated population. Descriptive statistics for demographic measures were calculated using SPSS 25 and are summarised in Table 1. Although there is no significant difference between the perceived economic level at the place of residence and the perceived economic level at the place of migration, it is in the middle-income range at 65% and 62.4%, respectively.

### 3.2. Scales

The survey was conducted with the participation of 700 people whose birthplace was the province of Artvin. The questions were grouped according to demographic characteristics, reasons for migration, social practices of the Artvin municipality, migration from Artvin, and reverse migration. The survey results were analysed with a reliability level of 95%. The frequency (*n*) and percentage (%) for categorical (qualitative) variables in the data obtained from the survey studies, the mean (X), the standard deviation (sd), and the minimum and maximum values for numerical (quantitative) variables were calculated. The Pearson correlation test was used to determine the factors influencing "migration" and "reverse migration" in Artvin Province. The test for independent groups and the ANOVA test were also used. Exploratory and confirmatory factor analyses were conducted for the validity of the scales, and their reliability was calculated [72].

A questionnaire was designed with a five-point Likert scale, where 1 stands for "strongly disagree" and 5 for "strongly agree". A total of 64 questions were asked to the participants. The questions were inspired by the literature but not copied verbatim. In a sense, they were designed and adapted to suit the nature of the research and to benefit from the literature. Nine different questions were asked to capture the demographic characteristics of the users. Questions were asked about the person, gender, age, education level, place of residence, time of migration to the place of residence, occupation, average monthly household income, and level of perceived income in the place of residence and city of migration. To explore the reasons for migrating from Artvin to the city of residence, eleven questions were asked about education, occupation, social life, income level, municipal services, climatic conditions, health, cleanliness of air, soil conditions, adaptation to the city, and level of satisfaction with the city (Table A1). Ten questions were asked about the social municipality and project practices of Artvin municipality (Table A2). Sixteen questions

were asked to measure the level of follow-up of traditional media and social media by those who migrated from Artvin (Table A3). Eighteen questions were asked to participants for research on returning to Artvin (Tables A4 and A5).

**Table 1.** Demographic characteristics of the research sample.

| *n* = 700 | | *n* | % |
|---|---|---|---|
| Gender | Male | 531 | 75.9 |
| | Female | 169 | 24.1 |
| Age | 18–24 | 67 | 9.6 |
| | 25–31 | 131 | 18.7 |
| | 32–38 | 136 | 19.4 |
| | 39–45 | 138 | 19.7 |
| | 46 and above | 228 | 32.6 |
| Education level | High school | 324 | 46.3 |
| | College | 115 | 16.4 |
| | Faculty | 188 | 26.9 |
| | Postgraduate (MSc) | 63 | 9.0 |
| | Doctorate (PhD) | 10 | 1.4 |
| Place of residence | Kocaeli | 101 | 14.4 |
| | Bursa | 156 | 22.3 |
| | İstanbul | 320 | 45.7 |
| | Ankara | 123 | 17.6 |
| Time of immigration to place of residence | 1–5 years ago | 67 | 9.6 |
| | 6–10 years ago | 62 | 8.9 |
| | 11–15 years ago | 66 | 9.4 |
| | 16–20 years ago | 101 | 14.4 |
| | 21 year and above | 404 | 57.7 |
| Occupation | Labour [2] | 166 | 23.7 |
| | Civil servant | 166 | 23.7 |
| | Farmer | 2 | 0.3 |
| | Self-employment | 124 | 17.7 |
| | Other | 242 | 34.6 |
| Average monthly household income [1] | Minimum wage (USD 355) and below | 25 | 3.6 |
| | Minimum wage–USD 422 | 77 | 11.0 |
| | USD 422–USD 528 | 102 | 14.6 |
| | USD 528–USD 704 | 166 | 23.7 |
| | USD 704 and above | 329 | 47.1 |
| Perceived economic level at place of residence | Low income | 206 | 29.4 |
| | Middle income | 455 | 65.0 |
| | High income | 39 | 5.6 |
| Perceived economic level at place of migration | Low income | 220 | 31.4 |
| | Middle income | 437 | 62.4 |
| | High income | 43 | 6.1 |
| Considering migrating back to Artvin | Yes | 409 | 58.4 |
| | No | 291 | 41.6 |

[1] Average monthly household income is calculated based on the average exchange rate in 2019 (USD 1 = TRY 5.68). Source: Ministry of Labour and Social Security, Republic of Turkey. [2] A person who produces in a workplace such as a factory, workshop, mine, agricultural enterprise, etc., using his/her body, head power, or both, in exchange for a certain wage.

### 3.3. Reliability and Validity of the Scales

To validate the measurement model, reliability as well as content convergence and distinctive validity must be assessed. Bagozzi et al. [73] state that convergent validity is achieved when all items that form the structure, i.e., the factors, are statistically significant. It is also suggested that convergent validity is accepted when the CR value exceeds 0.7,

even if the AVE value remains below 0.5 [74,75] (p. 141). Table 2 shows that the values for reliability, CR, and AVE agree with the recommended values, so that the reliability and convergent validity of all structures are verified. The Cronbach's alpha coefficient varies from 0 to 1. If the scale has a value of more than 0.60, it is very reliable or highly reliable according to the evaluation criteria [76]. SMA, personal SMU, SMU of Artvin municipality, and the traditional media have high reliability, and the SMC, migration, place of residence, and RM have very high reliability.

**Table 2.** Reliability analysis results of the scales.

|  | Cronbach's Alpha | AVE | CR |
|---|---|---|---|
| SMA | 0.636 | 0.756 | 0.949 |
| Personal SMU | 0.728 | 0.376 | 0.739 |
| Institutional SMU | 0.796 | 0.422 | 0.876 |
| SMU | 0.835 | 0.425 | 0.878 |
| Traditional SMU | 0.660 | 0.346 | 0.636 |
| RM | 0.802 | 0.304 | 0.791 |

## 4. Results

### 4.1. Correlation Analysis

Table 3 shows the correlations for the variables in the study. It was found that social municipality activities have a positive relationship with all the variables of the study. The results also show that the dependent variable, reverse migration, also has positive correlations with all the variables.

**Table 3.** Correlation matrix.

|  | SMA | Per. SMU | Ins. SMU | SMU | TMU | RM |
|---|---|---|---|---|---|---|
| SMA | 1 | 0.290 ** | 0.589 ** | 0.516 ** | 0.457 ** | 0.416 ** |
| Per. SMU |  | 1 | 0.451 ** | 0.852 ** | 0.442 ** | 0.267 ** |
| Ins. SMU |  |  | 1 | 0.851 ** | 0.623 ** | 0.416 ** |
| SMU |  |  |  | 1 | 0.625 ** | 0.399 ** |
| TMU |  |  |  |  | 1 | 0.316 ** |
| RM |  |  |  |  |  | 1 |

** $p < 0.01$.

### 4.2. Hypotheses Tests

The research hypotheses were tested using the software AMOS 16.0 by running two different structural equation models using the 2000 resampling option with the bootstrap technique. In the first model, the mediating role of social media use was tested. In the model, SMA was independent, RM was dependent, and ISMU and TMU were mediating variables. The model is plotted in Figure 2, and the test results are shown in Table 4.

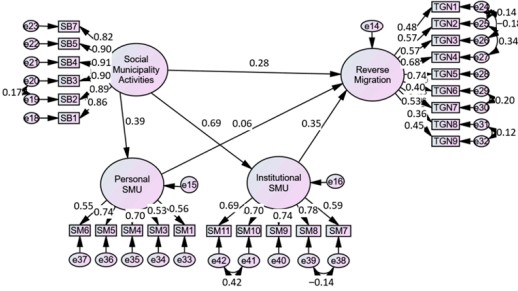

**Figure 2.** Social media use mediation model path diagram.

**Table 4.** Bootstrap results for social media usage mediation.

| Effects | Beta | Unstandardized SE | LLCI-ULLC |
|---|---|---|---|
| **Direct Impacts** | | | |
| SMA → RM | 0.151 *** | 0.046 | 0.070–0.249 |
| SMA → PSMU | 0.218 *** | 0.039 | 0.145–0.299 |
| SMA → ISMU | 0.428 *** | 0.046 | 0.345–0.528 |
| PSMU → RM | 0.056 | 0.079 | −0.072–0.242 |
| ISMU → RM | 0.305 *** | 0.098 | 0.137–0.521 |
| **Indirect Impact** | | | |
| SMA → ISMU (ISMU) → RM | 0.143 *** | 0.036 | 0.085–0.229 |

*** $p < 0.001$.

Table 4 shows that SMA (B = 0.151 ***, SE = 0.046, 95% CI = [0.070–0.249], $p < 0.05$) and ISMU (B = 0.305 ***, SE = 0.098, 95% CI = [0.137–0.521], $p < 0.05$) have a significant and positive effect on RM. However, the impact of PSMU (B = 0.056, SE = 0.079, 95% CI = [−0.072–0.242], $p > 0.05$) on RM is insignificant.

The results show that the indirect impact of SMA on RM is significant (B = 0.143 ***, SE = 0.036, 95% CI = [0.085–0.229], $p < 0.05$). Since the impact of PSMU on RM is insignificant, it can be said that ISMU acts as a mediator of the relationship between SMA and RM. The mediating impact in the model indicates a significant partial mediating role. Depending on these results, H1a, H1b, H3, H4, and H6b are supported, but H6a is not. The overall fit of the model was acceptable (CMIN/df = 2.426, GFI = 0.889, AGFI = 0.860, CFI = 0.933, RMSEA = 0.059, NFI = 0.892) [76,77].

In the second model, the mediating role of the use of traditional media was tested. In the model, SMA was independent, RM was dependent, and TMU was the mediating variable. The model is plotted in Figure 3, and the test results are shown in Table 5.

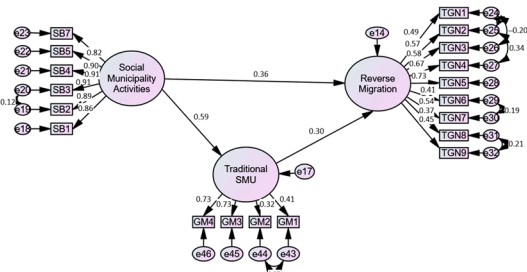

**Figure 3.** Traditional media use mediation model path diagram.

**Table 5.** Bootstrap results for traditional media usage mediation.

| Effects | Beta | Unstandardized SE | LLCI-ULLC |
|---|---|---|---|
| **Direct Impacts** | | | |
| SMA → RM | 0.199 *** | 0.044 | 0.118–0.287 |
| SMA → PSMU | 0.277 *** | 0.050 | 0.184–0.383 |
| SMA → ISMU | 0.360 *** | 0.140 | 0.161–0.731 |
| **Indirect Impact** | | | |
| SMA → TMU → RM | 0.100 *** | 0.032 | 0.049–0.179 |

*** $p < 0.001$.

Table 5 shows that SMA (B = 0.199 ***, SE = 0.044, 95% CI= [0.118–0.287], $p < 0.05$) and TMA (B = 0.360 ***, SE = 0.140, 95% CI= [0.161–0.731], $p < 0.05$) have a significant and positive effect on RM. The results show that the indirect impact of SMA on RM is significant (B = 0.100 ***, SE = 0.032, 95% CI= [0.049–0.179], $p < 0.05$). Thus, TMU is found to mediate the relationship between SMA and RM. The mediating impact in the model indicates a significant partial mediating role. Depending on these results, H2, H5, and H7 are supported. The overall fit of the model was acceptable (CMIN/df = 2.513, GFI = 0.916, AGFI = 0.886, CFI = 0.948, RMSEA = 0.061, NFI = 0.917).

## 5. Discussion

As SMAs increase, volunteers are using social media increasingly as they adopt these activities. H1b, on the other hand, uses social media to disseminate SMU activities. This use presents a task. For this reason, they use social media as part of the mission. In H3, social municipality activities in the city provide information to their relatives through social media activities in that city. As these activities are successful, they perform reverse migration. In other words, the desire to return increases. In H4, the high amount of news about the municipality on social media creates awareness among people, and through this awareness, they return to the city to which they migrated. Those who use social media personally in H6a have no impact on the point of return of immigrants, that is, individual posts on social media have no influence on reverse migration. One could argue that reverse migration has not been supported by the users of social media platforms. In H6b, the social activities of the municipality are communicated to the inhabitants of other cities through corporate media and have a mediating effect on the provision of reverse migration. Table 6 below summarizes the findings and conclusions of the study. From the table, all of them were supported except that Personal SMU plays a mediating role between SMA and RM.

**Table 6.** Hypothesis confirmation table

| | |
|---|---|
| **H1a:** *SMA has a positive impact on personal SMU.* | Supported |
| **H1b:** *SMA has a positive impact on institutional SMU.* | Supported |
| **H3:** *SMA has a positive impact on RM.* | Supported |
| **H4:** *SMU has a positive impact on RM.* | Supported |
| **H6a:** *Personal SMU plays a mediating role between SMA and RM.* | Not Supported |
| **H6b:** *Institutional SMU plays a mediating role between SMA and RM.* | Supported |

The activities of social municipality in Artvin, a province where there has been emigration, are shared by corporate social media users with their compatriots in other cities who have emigrated. In this way, they can have a mediating effect on their decisions to return to the cities to which they have emigrated. The results show that both the level of income and education and the age of the migrants were not taken into account in supporting return migration. While return migration is influenced by SMU and SMA, the cities of Ankara, Istanbul, Bursa, and Kocaeli, to which people migrated from Artvin, play a different role related to municipal activities in these cities. While the perception of reverse migration to Artvin is highest in the city of İstanbul, it is lowest in Ankara. The reason for this is that those who migrate to Ankara usually have job guarantees.

When analysing how many years ago people migrated from Artvin, the practices of the social municipality and the use of social media have no positive or negative influence on the period mentioned. People's occupations also have no influence on reverse migration. On the other hand, there is a correlation between those who think of returning to Artvin and those who do not, in relation to SMA. There is also a correlation between those who have plans to return to Artvin and those who are not thinking of disclosing reverse migration.

## 6. Conclusions

Artvin takes a different position in the migratory movements of Turkey. This is because, in terms of migratory movements, Artvin is among the top five provinces that people emigrate to in Turkey. Artvin is also in the destination region of international migration. It would not be difficult to call Artvin, which is located at the crossroads of internal and external migration, a city of migration. The migration experience of Artvin, which lies at the centre of different types of migration and different migration patterns, deserves to be the subject of a sociological study. Indeed, this was the main motivation for this study.

In general, the migration theories reviewed so far examine the causes of migration and focus on the quantitative and qualitative picture of migration. In addition, many migration theories are economically based and follow an approach that is detached from

the historical perspective. However, the migration phenomena of our time are so significant that they cannot be disconnected from the historical perspective in the context of their social background, and the temporal–spatial relationship of any migration that takes place cannot be ignored. Another visible trend in migration research is the shift from the local context to the international or transnational context. In this sense, broad studies are necessary for an international context, but they will be incomplete in images that are disconnected from the local context. Local studies are more complex and require open-ended fieldwork. These detailed field studies are vital for building larger images and are necessary to design longitudinal studies methodologically.

In this sense, idiosyncratic research on the local context should be further developed. Local mobility and intermediary institutions still need to be better understood. Although this study focuses specifically on cases in Turkey, the most remote province in the northeast of the country was chosen because it is one of the most challenging regions for migrant populations in Turkey. In addition to the question of whether or not social municipalism can provide an advantage to a geographical location with difficult economic conditions, the mediation factor of the media was clearly highlighted.

In conclusion, this study highlights the importance of the relationship between SMA and migration and studies on reverse migration in the context of SMU, especially institutional SMU. The province of Artvin represents a very special case in this context, which is presented below:

Empirical evidence consistently shows that both migration and reverse migration have an impact on SMA and that the SMU also has an impact on reverse migration. Finally, traditional media also have an influence on migration and reverse migration.

**Author Contributions:** Conceptualisation and methodology, M.K. and C.Y.; data curation and software, M.K. and H.Ş.; analysis: M.K. and M.S.; writing the first draft: M.K.; reviewing: C.Y. and M.S. All authors have read and agreed to the published version of the manuscript.

**Funding:** This research received no external funding.

**Institutional Review Board Statement:** This study was conducted in accordance with the Declaration of Helsinki and approved by the Ethics Committee of Near East University Social Sciences Ethics Committee (YDÜ/SB/2019/459 and 06.08.2019).

**Informed Consent Statement:** Informed consent was obtained from all subjects involved in the study.

**Data Availability Statement:** Data are contained within this article.

**Acknowledgments:** This study is derived from a doctoral dissertation titled as follows: "Examining The Impact of Social Municipality on Migration as Communication and Public Relations: The Case of Artvin Province". The authors would like to thank Bülent Evre and Mehtap Kara for their valuable contributions and criticisms.

**Conflicts of Interest:** The authors declare no conflict of interest.

## Appendix A

**Table A1.** Distribution of the degree of participation in terms of reasons for migration.

| Survey Question | 1 | 2 | 3 | 4 | 5 | Mean | Std. Dev. |
|---|---|---|---|---|---|---|---|
| I migrated to the city where I live to continue my education. | 15.0 | 30.9 | 13.1 | 24.9 | 16.1 | 2.96 | 1.34 |
| I migrated to the city where I live because of the job field. | 9.4 | 10.3 | 6.6 | 45.6 | 28.1 | 3.73 | 1.24 |
| I migrated to the city where I live for social life. | 15.0 | 33.6 | 12.6 | 29.4 | 9.4 | 2.85 | 1.26 |
| Because the income level of the city where I live is high. | 11.1 | 26.9 | 10.1 | 40.4 | 11.4 | 3.14 | 1.25 |
| I migrated to the city where I live due to Municipal Services. | 23.1 | 43.1 | 15.3 | 13.7 | 4.7 | 2.34 | 1.12 |

**Table A1.** *Cont.*

| Survey Question | 1 | 2 | 3 | 4 | 5 | Mean | Std. Dev. |
|---|---|---|---|---|---|---|---|
| I migrated to the city where I live due to climatic conditions. | 24.0 | 47.6 | 13.4 | 11.0 | 4.0 | 2.23 | 1.06 |
| I migrated to the city where I live for health reasons. | 19.0 | 44.6 | 11.6 | 19.1 | 5.7 | 2.48 | 1.17 |
| I migrated to the city where I live because of the cleanliness of the air. | 37.7 | 43.1 | 8.7 | 7.6 | 2.9 | 1.95 | 1.01 |
| I migrated to the city where I live because of the land conditions. | 25.0 | 41.3 | 12.7 | 16.6 | 4.4 | 2.34 | 1.15 |
| I believe that I have adapted sufficiently to the city I live in. | 5.0 | 14.1 | 10.6 | 53.3 | 17.0 | 3.63 | 1.08 |
| I am happy to be in the city where I live. | 8.1 | 21.1 | 12.3 | 44.9 | 13.6 | 3.35 | 1.19 |

1: Strongly disagree, …, 5: strongly agree.

**Table A2.** Distribution of degree of participation in social municipality practices and campaigns.

| Survey Question | 1 | 2 | 3 | 4 | 5 | Mean | Std. Dev. |
|---|---|---|---|---|---|---|---|
| Services for young people are sufficient. | 6.4 | 11.0 | 32.9 | 33.4 | 16.3 | 3.42 | 1.08 |
| Services for the elderly are sufficient. | 4.1 | 8.7 | 33.9 | 34.9 | 18.4 | 3.55 | 1.02 |
| Services for women are sufficient. | 4.6 | 10.1 | 36.3 | 31.1 | 17.9 | 3.48 | 1.04 |
| Their services to the poor are sufficient. | 5.3 | 8.9 | 34.4 | 33.1 | 18.3 | 3.50 | 1.05 |
| Services for individuals with special needs (Disabled) are sufficient. | 4.0 | 8.3 | 36.4 | 32.3 | 19.0 | 3.54 | 1.02 |
| Funeral and burial services are sufficient. | 3.0 | 4.6 | 28.3 | 39.3 | 24.9 | 3.78 | 0.97 |
| Cultural services are sufficient. | 4.4 | 8.7 | 28.1 | 37.9 | 20.9 | 3.62 | 1,05 |
| I find the social life areas he has created sufficient. | 3.0 | 8.3 | 22.9 | 40.9 | 25.0 | 3.77 | 1.01 |
| "Welcome baby application" for newborn babies and their families is sufficient. | 2.9 | 5.0 | 33.0 | 38.9 | 20.3 | 3.69 | 0.95 |
| I find compassionate service successful. | 2.7 | 4.9 | 31.7 | 40.1 | 20.6 | 3.71 | 0.94 |

1: Strongly disagree, …, 5: strongly agree.

**Table A3.** Distribution of participation in relation to aspects of media use.

| Survey Question | 1 | 2 | 3 | 4 | 5 | Mean | Std. Dev. |
|---|---|---|---|---|---|---|---|
| I regularly read the newspaper | 4.7 | 24.0 | 10.3 | 47.1 | 13.9 | 3.41 | 1.13 |
| I watch TV regularly | 3.7 | 21.0 | 6.4 | 56.3 | 12.6 | 3.53 | 1.07 |
| I am a regular social media (Facebook, Twitter, Instagram, YouTube etc.) user. | 2.1 | 10.3 | 4.6 | 56.4 | 26.6 | 3.95 | 0.96 |
| I regularly use Facebook the most. | 8.6 | 26.9 | 6.4 | 42.1 | 16.0 | 3.30 | 1.26 |
| I use Twitter the most regularly. | 15.3 | 35.9 | 10.9 | 27.6 | 10.4 | 2.82 | 1.28 |
| I regularly use Instagram the most. | 7.9 | 23.6 | 8.0 | 42.0 | 18.6 | 3.40 | 1.25 |
| I regularly use YouTube the most. | 6.7 | 29.4 | 7.3 | 44.4 | 12.1 | 3.26 | 1.20 |
| I regularly use other social media accounts the most. | 11.3 | 32.1 | 14.6 | 33.7 | 8.3 | 2.96 | 1.20 |
| I follow the works of Artvin Municipality mainly through traditional media. | 3.1 | 17.1 | 17.6 | 49.1 | 13.0 | 3.52 | 1.02 |
| I follow the works of Artvin Municipality most from my close circle. | 1.4 | 10.3 | 12.9 | 61.1 | 14.3 | 3.77 | 0.87 |
| I follow the works of Artvin Municipality mostly on social media. | 3.1 | 13.0 | 14.7 | 49.7 | 19.4 | 3.69 | 1.03 |
| I mostly follow the works of Artvin Municipality on the municipality's website. | 6.1 | 31.1 | 22.0 | 30.1 | 10.6 | 3.08 | 1.13 |
| I follow the works of Artvin Municipality mainly from the social media accounts of the municipality. | 4.9 | 20.3 | 19.0 | 42.4 | 13.4 | 3.39 | 1,10 |
| I follow the works of Artvin Municipality from local media | 6.0 | 24.0 | 21.7 | 37.7 | 10.6 | 3.23 | 1.11 |
| I find Artvin Municipality's website active enough | 4.9 | 16.0 | 39.9 | 29.6 | 9.7 | 3.23 | 0.99 |
| I find Artvin Municipality's social media accounts active enough | 3.9 | 14.1 | 36.3 | 34.3 | 11.4 | 3.35 | 0.99 |

1: Strongly disagree, …, 5: strongly agree.

**Table A4.** Distribution of participation in terms of reverse migration among those planning to return to Artvin.

| Survey Question | 1 | 2 | 3 | 4 | 5 | Mean | Std. Dev. |
|---|---|---|---|---|---|---|---|
| I'm considering going back for educational reasons | 14.8 | 48.4 | 16.3 | 14.1 | 6.4 | 2.49 | 1.10 |
| I'm thinking of going back because of the job site | 15.1 | 39.5 | 13.8 | 21.5 | 10.1 | 2.72 | 1.24 |
| I'm thinking of coming back for his social life | 6.2 | 21.8 | 8.4 | 42.9 | 20.6 | 3.50 | 1.21 |
| I'm thinking of going back because the income level is high | 14.1 | 38.8 | 22.0 | 18.3 | 6.9 | 2.65 | 1.14 |
| I'm thinking of going back because of Municipal Services | 9.6 | 27.1 | 23.6 | 24.9 | 14.8 | 3.08 | 1.22 |
| I'm thinking of going back due to the climatic conditions | 5.4 | 20.7 | 13.8 | 43.0 | 17.0 | 3.45 | 1.15 |
| I'm considering going back for health reasons | 6.9 | 31.8 | 16.5 | 33.5 | 11.3 | 3.11 | 1.17 |
| I'm thinking of going back because of the clean air | 4.7 | 3.2 | 3.9 | 39.7 | 48.5 | 4.24 | 1.01 |
| I'm thinking of going back because of the terrain conditions. | 7.6 | 29.8 | 17.2 | 31.3 | 14.0 | 3.14 | 1.21 |

1: Strongly disagree, ..., 5: strongly agree.

**Table A5.** Distribution of participation in reverse migration among those who do not plan to return to Artvin.

| Survey Question | 1 | 2 | 3 | 4 | 5 | Mean | Std. Dev. |
|---|---|---|---|---|---|---|---|
| I'm considering going back for educational reasons | 18.2 | 50.2 | 19.2 | 10.8 | 1.5 | 2.27 | 0.93 |
| I'm thinking of going back because of the job site | 20.2 | 49.8 | 18.2 | 9.4 | 2.5 | 2.24 | 0.96 |
| I'm thinking of coming back for his social life | 15.5 | 43.0 | 22.0 | 16.5 | 3.0 | 2.49 | 1.04 |
| I'm thinking of going back because the income level is high | 16.5 | 51.5 | 20.0 | 9.0 | 3.0 | 2.31 | 0.95 |
| I'm thinking of going back because of Municipal Services | 11.6 | 41.7 | 30.2 | 12.6 | 4.0 | 2.56 | 0.99 |
| I'm thinking of going back due to the climatic conditions | 9.5 | 41.0 | 21.0 | 24.0 | 4.5 | 2.73 | 1.07 |
| I'm considering going back for health reasons | 12.0 | 42.5 | 24.0 | 20.0 | 1.5 | 2.57 | 0.99 |
| I'm thinking of going back because of the clean air | 3.5 | 20.6 | 19.6 | 39.7 | 16.6 | 3.45 | 1.10 |
| I'm thinking of going back because of the terrain conditions. | 12.6 | 45.2 | 26.6 | 11.6 | 4.0 | 2.49 | 0.99 |

1: Strongly disagree, ..., 5: strongly agree.

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
