# Peer review of "The Mediation Effect of Media: Artvin, Reverse Migration, and Social Municipalism"

_sustainability, doi:10.3390/su151914304_

Round 1

Reviewer 1 Report

This is not my field, so it is possible I may be mistaken, but I feel like the Introduction section could be edited for brevity. Is it truly necessary to cover all of the concepts and with as much depth as there is in the current manuscript in order to understand the current research? I would encourage the authors to consider whether the Introduction could be more concise.

"The survey results were analysed with a confidence level 374 of 95% using the SPSS 25 programme. The frequency (n) and percentage (%) for categori- 375 cal (qualitative) variables in the data obtained from the survey studies, the mean (X), the 376 standard deviation (ss) and the minimum and maximum values for numerical (quantita- 377 tive) variables were calculated."

This is a summary of how you calculated values reported in Table 1, correct? I do not think it is necessary to describe the analysis of demographic descriptors. If you do, it should be under Participants rather than Scales (also this subsection heading of Scales is a bit confusing), and it should at most read, "Descriptive statistics for demographic measures were calculated using SPSS 25 and are summarized in Table 1."

"A questionnaire was designed with a five-point Likert scale, where 1 stands for “strongly disagree” and 5 for “strongly disagree”."

A questionnaire measuring what exactly? Most if not all of the variables you discuss measuring in this section do not seem to be measured on Likert-type scales.

"The questions were inspired by the literature but not copied verbatim"

Which previously validated measure was being adapted exactly and in which specific previous paper was that measure validated? Again, I could use more immediate detail on what was being measured. How was the current measure modified from its original version?

"The Cronbach's alpha coefficient varies from 0 to 1. According to the evaluation criteria, if the scale has a value of 0.00 < 0.40 it is 408 not reliable, if it has a value of 0.40 < 0.60 the reliability is low, if it has a value of 0.60 < 409 0.80 the scale is fairly reliable and if it has a value of 0.80 < 1.00 the scale is very reliable."

The authors need to get rid of this and all similar writing that over-explains concepts that are ubiquitous to science and / or statistics. It reads like a textbook. We need to assume that our readers know these things (even though we know that many of our readers who may be students do not know these things), otherwise every paper would need to describe significance values, power analyses, and Type I and II error rates and just about every other statistical concept you can think of. 

The authors need to separate the description of the Demographic measures from the other measures such as SMA, SMU, and RM. Demographic measures could simply be reported as I stated above. The other measures should each have their own subsections.

SMA. Report how this variable was measured. Name of the scale, number of items cronbach's alpha, example items, range of possible scores.

Then do the same thing for each of those variables one at a time.

I think I will reserve additional comments on the Results and Discussion until when (or if) I see a revision of the Introduction and Method. I do not want to make comments on those sections which may turn out to be irrelevant after I get a clearer understanding of the earlier sections.

I also think it would be good to put some additional effort into the improving the English language prose. 

Reviewer 2 Report

1. State the formula used to determination of sample size (line number 356).

2. Describe more about survey (e.g., when and who were conducted the survey and how many questions were in the questionnaire).

3. Did the authors did the Institutional Review Board (IRB)seek permission for this study?

4. Table 2: Replace "," by "."

5. Age, education, income, and gender were not considered as independent variables. These are important factors of migration/return migration. Why?

6. Study shortcomings are not mentioned in this manuscript. 

Minor editing of English language is required.

Reviewer 3 Report

Title - must be rephrased. It does not make any sense.

Introduction/ theoretical background - although it focuses on most of the concepts addressed in the paper, I still think this section should be more focused on two particular issues, namely return migration and social municipalism, as the latter is specific for Turkey. 

There should also be a separate section or subsection about Artvin, detailing why it is a deviant case (social and economic background that influence migration + the most important and visible social municipalism practices, focusing on activities and campaigns targeted toward migrants).

Methodology and results - it is not clear how the link between social municipalism and media influence on migrants` decision has been established through the questionnaire. Also, it is not clear where social municipalism influences migrants' decision - in the destination area or in the origin area? What type of information is presented in the media and how can it influence their decision? How exactly can social municipalism influence return migration? How does it change the push factors that initially made people leave Artvin?

Also, please check the document attached for further point-by-point observations.

There are a lot of long sentences that are difficult to understand. Also, quite often, the meaning is not clear.

Round 2

Reviewer 3 Report

The authors have improved the manuscript and considered all recommendations. Thank you.

I see no issues now.

Author Response

Dear referee, thank you for your valuable contributions and thoughts.
